# GMTRouter: Personalized LLM Router over Multi-turn User Interactions

## Abstract

Large Language Model (LLM) routing has demonstrated strong capability in balancing response quality with computational cost. As users exhibit diverse preferences, personalization has attracted increasing attention in LLM routing, since even identical queries may require different models to generate responses tailored to individual needs. However, existing approaches are not fully personalized and often fail to faithfully capture the complex interactions between specific users and LLMs. Moreover, user preference data is typically scarce, noisy, and inconsistent in format, which limits the effectiveness of methods that rely solely on user-specific data. To address these challenges, we propose *GMTRouter*, which represents multi-turn user–LLM interactions as a heterogeneous graph with four node types: user, LLM, query, and response, thereby maximally preserving the rich relational structure of the interaction. Through a tailored message-passing mechanism, *GMTRouter* learns to capture user preferences from few-shot data within a lightweight inductive graph learning framework, enabling effective personalization. Extensive experiments demonstrate that *GMTRouter* consistently outperforms the strongest baselines, achieving 0.9%–21.6% higher accuracy and 0.006–0.309 higher AUC across multiple datasets. More importantly, we further demonstrate that *GMTRouter* can adapt to new users and evolving preferences using only few-shot data, without extensive fine-tuning.

## 1 Introduction

With the rapid development of the field of Large Language Models (LLMs), an increasing number of models with varying sizes, computational costs, and domain expertise have become available (Singhal et al., 2023; Luo et al., 2022). This makes LLM routing particularly important, as it enables the recommendation of appropriate LLMs for diverse user queries while balancing response quality with computational cost (Šakota et al., 2024; Stripelis et al., 2024). Such routing techniques are increasingly adopted in modern LLMs, including GPT-5 (OpenAI, 2025). At the same time, as more users engage with LLM routing services, differences in individual preferences become increasingly prominent: even identical queries may require different models to generate responses tailored to each user (Li et al., 2024b; Salehi et al., 2024). Therefore, this paper aims to highlight a pressing research question: *Can we design a personalized routing framework that aligns LLM selection with individual user preferences based on their interaction histories?*

Existing research has proposed various architectures for LLM routing frameworks: FrugalGPT introduces a BERT-based router that determines whether to switch to a larger LLM (Chen et al., 2023), while C2MAB-V constructs a bandit-based router to balance exploration and exploitation when selecting an LLM (Dai et al., 2024). GraphRouter formulates routing as a node classification task over a graph of queries, tasks, and LLMs (Feng et al., 2024). However, existing methods largely overlook the importance of extracting structured preference information from users' interaction histories: they are not fully personalized and often fail to faithfully model multi-turn conversations between users and LLMs, which represent the most common form of user–LLM interaction in real-world scenarios (Zhang et al., 2025a; Li et al., 2025b). Moreover, in real-world scenarios, the preference data provided by a single user is typically scarce, noisy, and inconsistent in format (Escamocher et al., 2024; Li et al., 2024a). This makes it challenging for methods that rely solely on user-specific data to learn user profiles (Salemi et al., 2024a; Gao et al., 2024) or use such data as a retrieval source to support routing (Au et al., 2025), thereby limiting their effectiveness.

**Interaction History Table**

| User ID | Query | Selected LLM | Response | Feedback |
|---|---|---|---|---|
| **User 1** | [Turn 1]"Please Explain ... ?"
[Turn 2]"Can a Process ... ?" | **GPT-4-1106-Preview** | [Turn 1]"Exothermic and endothermic ..."
[Turn 2]"Yes, a process ..." | [Turn 1]"rating: 3.0"
[Turn 2]"rating: 4.5" |
| **User 2** | [Turn 1]"Compose a blog ..." | **Claude-V1** | [Turn 1]"Title: Aloha Spirit ..." | [Turn 1]"ranking: Claude-V1
> Koala-13B" |
| **User 2** | [Turn 1]"Compose a blog ..." | **Koala-13B** | [Turn 1]"Aloha, fellow travelers ! ..." | [Turn 1]"ranking: Claude-V1
> Koala-13B" |
| **User 3** | [Turn 1]"Compose an email ..."
[Turn 2]"Rewrite your ..." | **Vicuna-13B** | [Turn 1]"Subject: An Exciting ..."
[Turn 2]"Subject: A Gental ..." | [Turn 1]"response: Subject: Embrace ..."
[Turn 2]"response: Subject: Soaring to ..." |

Figure 1: **Multi-turn user-LLM Interaction History Table.** Each row captures a multi-turn interaction with associated user feedback. User feedback can take various forms, including ratings, rankings, and ground-truth responses.

To address these challenges, we introduce **GMTRouter** (**G**raph **M**ulti-**T**urn **Router**), a heterogeneous graph-based LLM router based on multi-turn user interactions for personalized LLM routing. GMTRouter first sensitively identifies key entities within the user–LLM interaction process: users, LLMs, queries, and responses. By modeling these entities as different types of nodes and encoding their textual information into node embeddings, it maximally preserves the semantic information from the original data. To faithfully model the relational structure of multi-turn user–LLM interactions, GMTRouter organizes these diverse node types into a heterogeneous graph that captures complex relational dependencies. Each single-turn interaction is treated as a fundamental unit, and a virtual node, referred to as a *turn node*, is introduced to aggregate local information within each interaction round. We further transform user preference feedback into node features, enabling preference information to propagate across the graph. Moreover, rather than training the model to directly extract specific user profiles from large historical datasets, GMTRouter employs a novel inductive graph training framework to **enhance the model's ability to capture user preferences from few-shot data**. This design allows effective test-time personalization even under sparse interaction histories, such as cold-start scenarios involving new users. In summary, our main contributions are as follows:

- To the best of our knowledge, we are among the first to introduce an personalized LLM routing task based on multi-turn user interactions, providing new insights for this rapidly growing research field.

- We propose a novel personalized LLM routing framework, which faithfully models multi-turn user–LLM interactions as a heterogeneous graph, and learns to capture user preferences from few-shot data within a lightweight inductive graph learning framework.

- Through experiments on four datasets spanning diverse tasks, GMTRouter consistently outperforms the strongest baselines, achieving 0.9%–21.6% higher accuracy and 0.006–0.309 higher AUC. Moreover, we demonstrate that our method can efficiently adapt to unseen users with only a few interaction examples, without requiring retraining.

## 2 PRELIMINARIES

### 2.1 TASK FORMULATION

We introduce the personalized LLM routing task in this section. We focus on the multi-turn interaction scenario between users and LLMs with feedback (Wang et al., 2023b; Shi et al., 2024). Within a dialogue session, a user repeatedly interacts with a LLM: in each turn, the user issues a query, the LLM provides a response, and the user in turn supplies a piece of feedback. Such feedback can take multiple forms, including: (1) scalar scores (e.g., numerical ratings), (Wang et al., 2023c; 2024b); (2) preference rankings (e.g., choosing among multiple responses), (Yang et al., 2024; Sun et al., 2025); (3) ground-truth responses (e.g., directly providing the correct answer) (Gao et al., 2024; Salemi et al., 2024a). We structure these interactions into an **Interaction History Table**, illustrated in Figure 1, where each entry records the user ID, the selected LLM, the multi-turn queries and

Table 1: **The consistency of LLM preferences between users is significantly lower than the consistency within a single user's preferences.** The self-spearman score is substantially higher than the spearman scores computed across different users.

| Metric | Self Spearman | Global Spearman | Intra-cluster Spearman | Inter-cluster Spearman |
|---|---|---|---|---|
| Value | 0.7934 | 0.5239 | 0.5734 | 0.4424 |
| Percent | 100% | 65.99% | 72.28% | 55.74% |

generated responses, and the corresponding user feedback, thereby maximally preserving the rich relational infomation of the interaction.

Our personalized LLM routing task is then modeled as follows: Given $m$ users $\{u_1, \ldots, u_m\}$ and $n$ LLM candidates $\{m_1, \ldots, m_n\}$, as well as their historical interaction records:

$$\mathcal{H} = \{(u_i, m_i, \{(q^{(t)}, r^{(t)}, f^{(t)})\}_{t=1}^{T_i})\},$$

where $u_i$ is the user, $m_i$ is the selected LLM, and each record contains a multi-turn sequence of queries $q^{(t)}$, responses $r^{(t)}$, and feedback $f^{(t)}$ for $t = 1, \ldots, T_i$. When a user $u$ raises a new query $q$, the router is required to select an LLM $m \in \{m_1, \ldots, m_n\}$ to generate a response $r$ that best aligns with the user preferences, which is measured through the feedback $f$ provided by the user.

## 2.2 MOTIVATION

In this section, we highlight the significant differences in LLM preferences across users in the real world (Chevi et al., 2025; Wang et al., 2024a), emphasizing the importance of personalized LLM routing for enhancing user experience. We use the Chat-Bot Arena dataset (Chiang et al., 2024) to illustrate our findings, which contains extensive multi-turn conversations from numerous users, with pairwise human preference labels between various LLMs, enabling the study of real-world user–LLM interactions. From this dataset, we select 10 active users, each with at least 50 records, for detailed analysis. For each user, we randomly split their data into two halves and compute the win rates of each LLM within each half. We use Spearman correlation to quantify the consistency of preference rankings over LLMs (De Winter et al., 2016; Hauke & Kossowski, 2011). We then compute the Spearman correlation between the two halves to quantify their self-consistency in preferences over LLMs (Chevi et al., 2025; Jiang et al., 2025), reporting the average as a baseline for comparison with inter-user prefer-

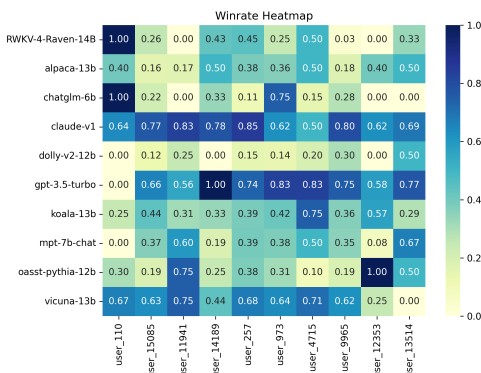

Figure 2: **Significant differences exist in LLM preferences across users.** The figure shows a heatmap of win rates for the 10 most popular LLMs across 10 active users in Chat-Bot Arena. The uneven color intensity within each row visually highlights the pronounced preference differences between users.

ence consistency. Next, based on the similarity of queries in each user's interaction history, we cluster users into three groups (Zeng et al., 2024; Li et al., 2025a), and compute pairwise Spearman correlation scores among users globally, within clusters, and across clusters (Cavallo, 2019; De Winter et al., 2016), reporting the corresponding averages as summarized in Table 1. We observe that global consistency in LLM preferences among users is substantially lower than individual self-consistency, reaching only 65.99% of the latter. Even within the same cluster, the Spearman score is only 72.28% of the self-consistency, highlighting the diversity of user preferences toward LLMs (Sun et al., 2025; Salemi et al., 2024a). To further visualize these differences, we select the 10 most frequently used models across these 10 users and present a win-rate heatmap in Figure 2, offering an intuitive depiction of the variability in user preferences. Therefore, to address the substantial inconsistency of LLM preferences across users, we propose **GMTRouter**, a framework that enables the personalized recommendation of suitable LLMs tailored to each user's individual preferences.

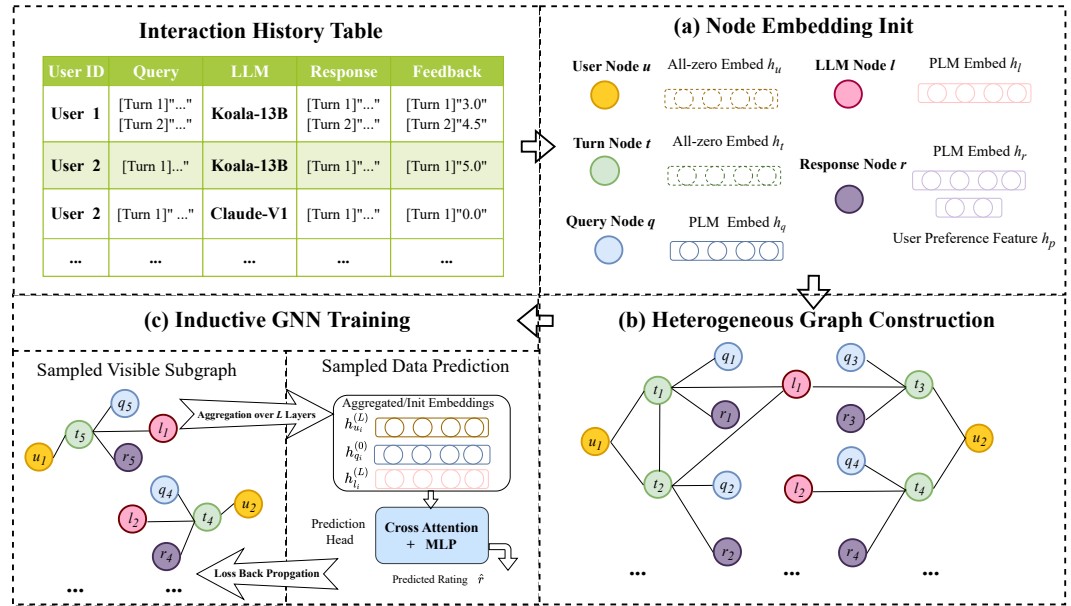

Figure 3: **Overview of GMTRouter.** (a) GMTRouter first extracts key entities: users, LLMs, queries, responses, and feedback, from the Interaction History Table and encodes their textual information using a PLM. (b) It then organizes these entities into a heterogeneous graph to faithfully model the relational structure of user–LLM interactions. (c) Within a lightweight inductive graph learning framework, GMTRouter learns to capture user preferences from few-shot data.

## 3 GMTROUTER: ROUTER OVER MULTI-TURN USER INTERACTIONS

**Method Overview**   As shown in Figure 3, GMTROUTER operates in three stages: (a) It first identifies the key entities in the Interaction History Table—users, LLMs, queries, responses, and feedback—modeling them as nodes and encoding the textual information into node embeddings to maximally preserve the information of the interaction process. (b) Based on the relational structure of user–LLM interactions, these nodes are connected to form a heterogeneous graph, which captures rich relational dependencies. To facilitate information propagation, we further introduce a virtual *turn node* that aggregates the information within each single-round interaction. (c) Finally, we adopt a novel inductive graph training framework to learn how to capture user preferences from few-shot data, thereby enhancing the model's ability to personalize under sparse user interaction histories.

### 3.1 NODE EMBEDDINGS INITIALIZATION.

First, our framework focuses on comprehensively extracting the information of various entities involved in the user–LLM interaction process from the Interaction History Table, along with their relational structures. As illustrated in part (a) of Figure 3, we extract four types of entities: user $u$, LLM $m$, query $q$, and response $r$, and formalize them as four corresponding node types. Their textual information is encoded using a pretrained language model (PLM) to obtain the initial node embeddings (Wang et al., 2022; 2023a), thereby preserving the semantic information from the original data. Specifically, we encode the query and response texts as their initial embeddings, denoted as $h_q$ and $h_r$. In addition, we transform various forms of user feedback in the Interaction History Table into numerical ratings and project them into a User Preference Feature $h_p$, which serves as another attribute on the response nodes. Concretely, ranking feedback is discretized into numerical ratings to ensure that higher-ranked responses receive higher scores (Banditwattanawong & Masdisornchote, 2025); for ground-truth response feedback, we compute the geometric distance between the embeddings of the ground-truth and the generated response as the rating criterion (Salemi et al., 2024a). For LLM nodes, instead of simply using their names or IDs (Ding et al., 2024; Chen et al., 2023), we encode the model overviews provided by AI/ML API platforms[1] as their node embeddings $h_m$,

---
[1]https://aimlapi.com/models/

which typically include key information such as model size, usage cost, and domain-specific capabilities, thereby enriching the node embeddings with important background knowledge. Finally, for user nodes, we do not assume the existence of text-based user profiles, as such information is often scarce and noisy in real-world applications (Su et al., 2024; Alzubaidi et al., 2023); therefore, we initialize user embeddings $h_u$ as zero vectors.

## 3.2 HETEROGENEOUS GRAPH CONSTRUCTION.

Next, we organize these nodes into a heterogeneous graph to model the relational structure of user–LLM interactions (Zhang et al., 2025b; Schlichtkrull et al., 2017). We consider each single-round user–LLM interaction as a fundamental unit and introduce a kind of virtual node, *the turn node*, to aggregate the information within each interaction round. As illustrated in part (b) of Figure 3, within each interaction round, the associated user node, LLM node, and generated query node, response node are all connected to the corresponding turn node, which serves to aggregate information from that round. For multi-turn conversations, the turn nodes corresponding to each round are sequentially connected in dialogue order, facilitating information propagation across turns. The turn node embedding $h_t$ is initialized as zero vectors. The resulting heterogeneous graph captures the rich relational dependencies inherent in user–LLM interactions, where turn nodes aggregate local information within each dialogue round and propagate it to user nodes, thereby facilitating the global aggregation of user preference information.

## 3.3 GNN AGGREGATION AND INDUCTIVE TRAINING

After constructing the user–LLM interaction histories into a heterogeneous graph, we train our GNN model on it. Notably, GMTRouter is a **general framework** that can incorporate any heterogeneous GNN as its backbone. We denote the GNN backbone used in our method as *GNN* throughout the rest of the paper.

To address scenarios with sparse user history (Su et al., 2024), instead of training the model to extract user profiles from large amounts of historical data (Lin et al., 2021; Wang et al., 2025), We employ an inductive framework along with **user-conditioned graph sampling** during training, enabling GMTRouter to **capture a user's preferences from only a few interaction records.**

**User-conditioned Graph Sampling**  As illustrated in the left of (c) in Figure 3, during each training epoch, we sample $k$ interaction histories for each user to construct a visible subgraph from the heterogeneous graph for message passing, and further sample data outside the visible subgraph as the prediction targets. We then use only these small sampled visible subgraphs and perform message aggregation separately for each user to update the node embeddings. Formally, at each layer $l$, the embedding of a node $v$ is updated by aggregating messages from its neighbors according to the message-passing mechanism of the backbone GNN:

$$h_v^{(l)} = \text{Norm}\left(\text{Dropout}\left(GNN^{(l)}(h_v^{(l-1)}, \mathcal{G}_{\text{sub}})\right)\right) \tag{1}$$

where $h_v^{(l)}$ denotes the embedding of node $v$ at layer $l$, and $\mathcal{G}_{\text{sub}}$ denotes the sampled visible subgraph. Norm($\cdot$) denotes layer normalization, and Dropout($\cdot$) is applied for regularization.

This restriction on the amount of data involved in message passing encourages the model to learn how to infer user preferences from very limited signals and to generalize efficiently to new users.

**LLM Routing with a Prediction Head.**  After completing $L$ layers of message aggregation, we obtain the updated node representations $h^{(L)}$. We then employ a **Prediction Head** module $f_{\text{pred}}$ for preference prediction. As illustrated in the right of (c) in Figure 3, the Prediction Head takes the updated user embedding $h_u^{(L)}$, LLM embedding $h_m^{(L)}$, and the query embedding $h^{(0)}q$ from PLM as input. It applies a cross-attention module, where the LLM embedding attends to the fused user–query context to extract relevant preference signals. The module outputs a scalar score $s_{u,q,m}$ for each LLM candidate, representing the likelihood that user $u$ would prefer $m$ to answer query:

$$s_{u,q,m} = f_{\text{pred}}(h_u^{(L)}, h_q^{(0)}, h_m^{(L)}) \tag{2}$$

These scores are then used to rank LLM candidates under the same $(u, q)$ condition. We normalize both the predicted scores and the ground-truth ratings, and apply a criterion function to compute the training loss, which is subsequently used to update the model parameters.

During inference, when a user raises a new query, we first sample $k$ interaction histories of that user to construct the visible subgraph and update the node embeddings. Then, the LLM candidate is selected from the candidate set $\mathcal{M}$ as the one with the highest predicted score:

$$m^* = \arg \max_{m \in \mathcal{M}} f_{\text{pred}}(h_u^{(L)}, h_q^{(0)}, h_m^{(L)}) \tag{3}$$

## 4 EXPERIMENT SETUP

### 4.1 DATASETS AND DATA PROCESSING

We conduct experiments on one real-world dataset and three additional synthetic datasets, covering four distinct tasks to enable a comprehensive evaluation of our approach.

- **Chatbot Arena (Chiang et al., 2024):** As mentioned in Section 2.2, we use the Chatbot Arena dataset to evaluate the personalized performance of our approach compared to baselines under authentic human preferences. For our experiments, we select the 11 users and 16 LLMs with the largest number of interactions. Detailed statistics are provided in Appendix B.1.
- **MT-Bench (Zheng et al., 2023):** MT-Bench is a benchmark for evaluating the reasoning and multi-turn conversational capabilities of LLMs, containing 80 multi-turn questions.
- **GSM8K (Cobbe et al., 2021):** GSM8K is a dataset of grade school-level math word problems, designed to assess LLMs' mathematical reasoning and problem-solving skills.
- **MMLU (Hendrycks et al., 2021a;b):** MMLU is a comprehensive benchmark covering 57 subjects from professional domains, used to measure general knowledge and multi-domain reasoning abilities of LLMs. We sample 10 questions from each subject for our experiments.
- **LaMP (Salemi et al., 2024b):** LaMP is designed to evaluate language models across multiple dimensions of personalization. We select the "Personalized Scholarly Title Generation" task, which provides pairs of paper titles and abstracts for multiple users and requires predicting the title a user would prefer given an abstract. We convert this task into a personalized routing dataset, with processing details provided in Appendix B.3.

**Data Processing**  For ChatBot Arena, we discretize the pairwise preferences to serve as the ratings for responses. For the other datasets, we adopt the data collected in Ong et al. (2024), which generated responses to all questions using "GPT-4-1106-preview" (Achiam et al., 2023) and "Mixtral-8x7B-Instruct-v0.1" (Jiang et al., 2024), and employed GPT-4 to provide quality annotations for open-ended questions. Based on this, we convert these datasets into multi-user personalized datasets. Specifically, for each response, we consider the following four dimensions: (a) Quality: For open-ended questions, we use the GPT-4 scores provided by Ong et al. (2024); for objective questions, we directly evaluate the correctness. (b) Cost: We calculate the cost of generating each response based on the API pricing provided by AI/ML API platform. (c) Response Length: We compute the token length of each response using the Contriever tokenizer (Izacard et al., 2021). (d) Rare Words: We count the number of rare words in each response using the *wordfreq* package (Speer, 2022).

We obtain the final rating of a response by computing a weighted sum of these four metrics. Different users are assigned different weightings to reflect their individual preferences over these dimensions (Feng et al., 2024; 2025). The specific weights used are provided in Appendix B.2.

**Data Splitting**  For all datasets, we partition the data into training, validation, and test sets with a 7:1:2 ratio, ensuring that users are evenly distributed across the splits. For the GMTRouter, we further adopt an additional splitting strategy: we sample 30% of the users and restrict their data to the test sets only, in order to evaluate the generalization ability of our method to new users unseen during training.

### 4.2 BASELINES

We compare our GMTRouter against the following baselines:

**Prompt-based: (1) Vanilla LLM.** We incorporate the query and the descriptions of candidate LLMs into the prompt, and feed it into LLaMA-3.1-70B (Grattafiori et al., 2024) to select the LLM. **(2) Personalized LLM.** Building on the Vanilla LLM, we retrieve from the training set the ten interaction histories most relevant to the user's query and incorporate them into the prompt. Leveraging in-context learning Dong et al. (2022), the LLM is then guided to perform personalized routing.

**Representative Router: (3) GraphRouter. (Feng et al., 2024)** We adopt GraphRouter as the representative router baseline. It is a graph-based model that formulates routing as a node classification task over a graph of queries, tasks, and LLMs with learned edge interactions, and has shown superior performance over many existing routers (Ding et al., 2024; Chen et al., 2023; Dai et al., 2024) in non-personalized settings. **(4) FrugalGPT (Chen et al., 2023)** utilizes a PLM to predict the score of the generation result of all LLMs given a query, and then selects the LLM with the highest score within a given cost. **(5) RouteLLM (Ong et al., 2024).** Learns to route queries among a weak-strong pair of LLMs. Following the official setup, we designate the weak model as the one with the lower average win rate in the dataset, and the strong model as the one with the higher win rate.

**Sequential / memory-based recommender: (6) MA-GNN (Chen Ma, 2020).** A memory-augmented GNN that models both *short-term* and *long-term* user interests through item-level message passing and a dedicated memory module. We treat each user's interaction history as a sequence of (query, LLM, feedback) records, where the sequential component captures short-term preference shifts and the memory module aggregates long-term preference signals; MA-GNN then predicts the preferred LLM for the current query. **(7) TIGER (Shashank Rajput, 2023).** A generative retrieval–based sequential recommender that models item sequences via semantic discrete codes; we regard LLMs as items and train TIGER to generate the semantic code of the best LLM conditioned on the user's past interactions and current query, ranking candidate LLMs by their predicted likelihood.

### 4.3 METRICS

We evaluate the performance of all methods using two metrics:

- **Accuracy** measures how often the model correctly identifies the most preferred LLM to answer a given query from a specific user.
- **AUC-ROC** (Area Under the Receiver Operating Characteristic Curve) measures the model's ability to correctly rank candidate LLMs according to user preferences. We employ a pairwise approach (C-index), defined as the probability that the predicted score $s_+$ for a preferred response (higher rating, $r_+$) is greater than the score $s_-$ for a less preferred response (lower rating, $r_-$) over all comparable pairs: $\text{AUC} = \Pr(s_+ > s_-) + \frac{1}{2}\Pr(s_+ = s_-)$.

### 4.4 IMPLEMENTATION DETAILS

We implement our method using PyTorch Geometric (Fey & Lenssen, 2019) and conduct all experiments on a single NVIDIA RTX A6000 GPU. We employ Contriever (Izacard et al., 2021) as the PLM to obtain the initial node embeddings. We adopt the Heterogeneous Graph Transformer (HGT) (Ziniu Hu, 2020) as our model backbone due to its strong capability to maintain dedicated representations for different types of nodes. Additionally, we experiment with various other heterogeneous GNNs as the backbone to investigate their impact on GMTRouter's performance. Experimental details are provided in Appendix F.4. We set the visible data size per user to $k = 10$ during both training and inference and adopt Entropy Loss as our loss function. In Section 5.2, we will experimentally analyze the impact of different values of $k$ on our method, and hyperparameter details are provided in Appendix A.1.

## 5 EXPERIMENT RESULTS

### 5.1 COMPARISON WITH BASELINES

We compare GMTRouter with baselines across four datasets in Table 2. We observe that GMTRouter consistently outperforms all baselines, delivering an improvement of 0.9%–21.6% on accuracy and 0.006–0.309 on AUC compared to the strongest baselines, demonstrating the superior-

Table 2: **GMTRouter consistently outperforms baselines across all datasets.** Bold and underline denote the best and second-best results. The results are averaged over multiple runs. Since RouteLLM and FrugalGPT are inherently binary routers, we evaluated them only in the binary setting from our datasets.

| Method | Chatbot-Arena | | MT-Bench | | GSM8K | | MMLU | | LaMP | |
|---|---|---|---|---|---|---|---|---|---|---|
| | ACC | AUC | ACC | AUC | ACC | AUC | ACC | AUC | ACC | AUC |
| Vanilla LLM | 0.525 | 0.741 | 0.481 | 0.457 | 0.546 | 0.533 | 0.473 | 0.475 | - | - |
| Personalized LLM | 0.646 | 0.780 | 0.437 | 0.491 | 0.553 | 0.536 | 0.675 | 0.678 | 0.312 | 0.605 |
| GraphRouter | 0.771 | 0.869 | 0.568 | 0.550 | 0.717 | 0.792 | 0.699 | 0.746 | 0.345 | 0.652 |
| FrugalGPT | 0.562 | 0.622 | 0.551 | 0.552 | 0.504 | 0.515 | 0.545 | 0.575 | - | - |
| MA-GNN | 0.673 | 0.775 | 0.679 | 0.739 | 0.636 | 0.648 | 0.702 | 0.758 | 0.347 | 0.661 |
| TIGER | 0.739 | 0.735 | 0.656 | 0.691 | 0.639 | 0.683 | 0.710 | 0.764 | 0.339 | **0.698** |
| RouteLLM | 0.492 | 0.485 | 0.480 | 0.475 | 0.499 | 0.498 | 0.532 | 0.513 | - | - |
| Ours | 0.774 | **0.875** | **0.784** | **0.859** | **0.773** | **0.859** | **0.771** | **0.870** | **0.349** | 0.662 |
| Ours (new user) | **0.780** | 0.858 | 0.759 | 0.824 | 0.756 | 0.833 | 0.751 | 0.831 | - | - |

Table 3: **GMTRouter requires only minimal storage and GPU resources.**

| HGT Params | Pred Head Params | Total Params | Storage Overhead | Max GPU Usage |
|---|---|---|---|---|
| 26.6M | 0.85M | 27.4M | 109.6MB | 4.3GB |

ity of our framework. For Personalized LLM, although incorporating user interaction histories into prompts leads to improvements over Vanilla LLM on most datasets, it still lags behind GMTRouter by at least 9.6% in accuracy and 0.095 in AUC. This highlights the limited ability of LLMs to extract preference patterns from noisy user data. Moreover, our method consistently outperforms GraphRouter, a representative router that has shown strong performance in non-personalized LLM routing tasks, across all datasets. These results validate the importance of leveraging structured information from multi-turn user–LLM interaction data, together with user preference signals, to better align LLM selection with diverse user needs. Furthermore, even when 30% of users are not present in the training set, our method achieves performance comparable to the standard setting, underscoring its strong generalization ability to new users.

**Our Framework is Lightweight** We report the parameter count, storage overhead, and training resource requirements of GMTRouter in Table 3. With only 27.4M trainable parameters and a 109.6MB model size, our framework remains compact compared to existing routing models. During training, only 4.3GB of GPU memory is needed, making it feasible to train on a single modern GPU without specialized hardware.

## 5.2 CASE STUDIES

**Investigating the Impact of Visible Data Size $k$** We investigate the impact of $k$ visible data per user on the quality of the aggregated node embeddings. The results on GSM8K and MMLU are shown in Figure 4. As $k$ increases, both accuracy and AUC improve, but beyond $k=10$, the performance begins to plateau or slightly decline, indicating diminishing returns from including additional visible data. This may be due to reduced generalization or potential instability caused by excessively large batch sizes during training (Keskar et al., 2016; Oyedotun et al., 2022). Therefore, we choose $k=10$ as a balanced setting for capturing user preferences without compromising generalization.

**Generalization to New Users** We further investigate the personalized capability of our method in few-shot scenarios with new users. Specifically, we evaluate on the GSM8K and MMLU by sampling 30% users from each dataset and varying the number of visible data $k \in \{3, 5, 8, 10, 15, 20\}$. Figure 5 presents averaged results of the sampled users under two settings: (i) the old user setting, where their records are included in the training set, and (ii) the new user setting, where they appear only in the validation and test sets. We observe that new users achieve results comparable to old users, and their performance curves consistently peak far above the GraphRouter baseline. These findings demonstrate that **our approach effectively learns to capture user preferences from few-shot data and can adapt to new users without requiring extensive fine-tuning.**

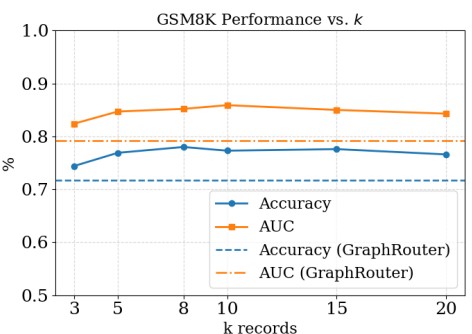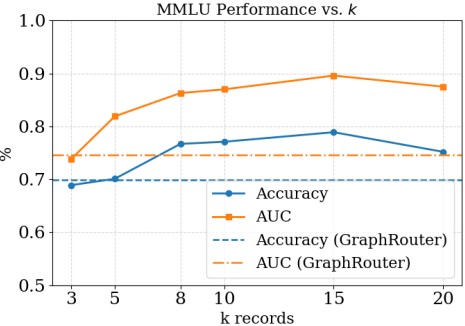

Figure 4: This figure illustrates **the impact of the visible data size** $k$ **on GMTRouter for GSM8K (left) and MMLU (right).** The dashed line represents the GraphRouter baseline. As $k$ increases, the performance of our method improves, but it saturates once $k$ reaches 10.

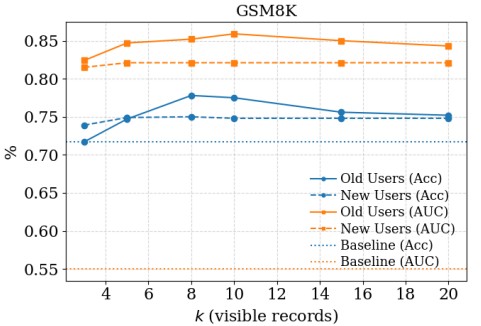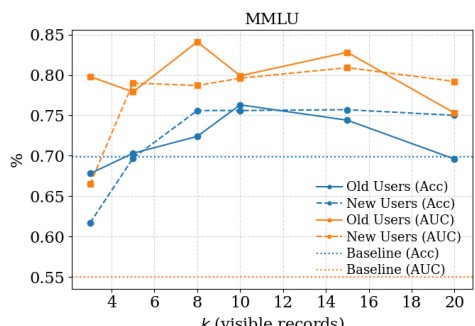

Figure 5: **This figure illustrates the result comparison between old-user and new-user settings for GSM8K (left) and MMLU (right).** The dashed line represents the GraphRouter baseline. The personalized performance under the new-user setting is comparable to that under the old-user setting, highlighting the strong generalization capability of our method.

### 5.3 ABLATION STUDIES

To evaluate the effectiveness of each design component of the GMTRouter, we conduct ablation studies along the following aspects.

- **w/o User Preference Feature** To verify the effectiveness of the user preference feature in propagating preference signals during GNN aggregation, we remove this feature in this variant. As a result, node embeddings are updated without incorporating preference ratings, which are used solely as supervision signals during training.

- **Dot-product Prediction Head** To evaluate whether the cross-attention prediction head captures non-linear interactions more effectively than standard similarity scoring when predicting the optimal model, we replace it in this variant with a simple dot product between the (user + query) and LLM embeddings.

- **Homogeneous Graph** To evaluate the effectiveness of our heterogeneous graph in capturing complex relationships among different entities in user–LLM interactions, we replace HGT with a homogeneous GNN, GraphSAGE (Hamilton et al., 2017), as the model backbone in this variant.

- **w/o User Embedding** To evaluate the effectiveness of user embeddings aggregated from the sampled visible graph for personalized prediction, we replace the user embeddings fed into the prediction head with zero vectors in this variant, thereby ablating their influence on the predictions.

The results of our ablation studies are presented in Table 4. As shown, our GMTRouter achieves the best performance on most metrics across all four datasets compared to the other variants, confirming the effectiveness of our design choices.

Table 4: **Ablation of design components.** We compare the full model with four variants: (1) removing the user preference features, (2) replacing the prediction head with a dot-product, (3) replacing HGT with GraphSAGE, (4) not using user embeddings during prediction. The best and second-best results are highlighted in **bold** and underline, respectively.

| Method | Chatbot-Arena | | MT-Bench | | GSM8K | | MMLU | |
|---|---|---|---|---|---|---|---|---|
| | Accuracy | AUC | Accuracy | AUC | Accuracy | AUC | Accuracy | AUC |
| w/o $h_p$ | 0.768 | 0.872 | 0.569 | 0.507 | 0.715 | 0.784 | 0.494 | 0.613 |
| Dot-product | **0.777** | 0.868 | 0.730 | 0.795 | 0.629 | 0.724 | 0.681 | 0.746 |
| Homo Graph | 0.768 | 0.873 | 0.569 | 0.645 | 0.635 | 0.648 | 0.494 | 0.487 |
| w/o $h_u$ | 0.771 | 0.873 | 0.569 | 0.631 | 0.725 | 0.814 | 0.701 | 0.771 |
| **GMTRouter** | 0.774 | **0.875** | **0.784** | **0.859** | **0.773** | **0.859** | **0.771** | **0.870** |

## 6 ADDITIONAL RELATED WORKS

**LLM Routing.** LLM routing focuses on enhancing inference efficiency and response quality by assigning queries to the most appropriate model (Yue et al., 2025; Zhang et al., 2025c). Recent work frames routing as learning with cost–quality tradeoffs (Kadavath et al., 2022; Dekoninck et al., 2024): RouteLLM learns from preference data Ong et al. (2024), and RouterBench offers standardized routing benchmarks Hu et al. (2024). BEST-Route jointly selects LLM and generation count at test-time via a bandit controller Ding et al. (2025). However, existing approaches are not fully personalized and fail to exploit user information from interaction histories as well as the structure of multi-turn dialogues.

**Heterogeneous Graph Learning.** HetGNNs are designed to model heterogeneous graphs by capturing complex multi-type interactions among various nodes and edges (Chien et al., 2021; Feng et al., 2019). HAN uses hierarchical attention over metapaths Wang et al. (2019), while MAGNN and HeCo improve metapath aggregation and cross-view contrast Fu et al. (2020); Wang et al. (2021). Transformers such as HGT provide inductive, relation-aware message passing with temporal encoding Ziniu Hu (2020). This enables rich relational structures in user–LLM interactions while leveraging inductive training to enhance generalization on sparse data from new users.

**Personalized LLMs.** Personalized LLMs adapt a fixed base model rather than selecting among models. Memory-style methods extend long-term user/context memory (M+), combine episodic and semantic traces (PRIME), or tune user-specific knowledge graphs from feedback (KGT) (Yu Wang, 2025; Xinliang Frederick Zhang, 2025; Jingwei Sun, 2024), while training-free patches port personalization across evolving bases (PortLLM) (Rana Muhammad Shahroz Khan, 2025). By contrast, we study *personalized routing*—per-user selection among candidate LLMs from multi-turn histories.

## 7 CONCLUSION

In this work, we introduced GMTRouter, a heterogeneous graph-based framework for personalized LLM routing. By modeling multi-turn user–LLM interactions as a heterogeneous graph and propagating preference signals across node types, our method effectively captures user-specific patterns even from few-shot, noisy data. Experiments across four benchmarks confirm that GMTRouter consistently surpasses strong baselines in both accuracy and AUC, while adapting efficiently to new users without retraining. These results highlight the value of structured interaction modeling for advancing preference-aware LLM routing and point to promising future directions in scalable, user-aligned LLM deployment.

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

# A    IMPLEMENTATION DETAILS

## A.1    MODEL CONFIGURATION AND HYPERPARAMETERS

**Architecture.**    We use a heterogeneous graph transformer (HGT) with:

- **GNN**: 2 layers (single-turn) or 3 (multi-turn), 768-dim hidden, 4-head HGTConv, Layer-Norm, dropout 0.1.

- **Predictor**: 4-head MLP with hidden dim 256, dropout 0.1; uses cross-attention where LLM attends to user+query.

**Training.**

- **Epochs**: 350    **LR**: 5e-4

- **Visible records/user** ($k$): {3, 5, 8, 10, 15, 20}

- **Batch size**: 256 supervision triplets

- **Ranking Objective**: prioritize AUC, then Accuracy

## A.2    TRAINING OF **GMTROUTER**

---

**Algorithm 1:** Training **GMTRouter**

---

**Data:** $\mathcal{D}_{\text{train}} = \{(x, y)\}$
**Hyperparams:** epochs $E$, visible $k$, supervision $s$, PLM, GNN $f_\phi$, predictor Pred
**Init:** PLM-encode all nodes; initialize node/edge features

1 **for** $e \leftarrow 1$ **to** $E$ **do**
2     $\mathcal{G}^{(e)} \leftarrow$ subgraph from $k|\mathcal{U}|$ visible records
3     $\mathcal{M}^{(e)} \leftarrow s$ held-out triples $(u, q, m)$
4     $h \leftarrow f_\phi(\mathcal{G}^{(e)})$             // message passing
5     **for** $(u, q, m) \in \mathcal{M}^{(e)}$ **do**
6        $\hat{y} \leftarrow \text{Pred}(h_u, q, h_m)$
7     Update $f_\phi$ and Pred by minimizing $\mathcal{L}_{\text{rank}}(\hat{y}, y)$

---

# B    DATASET PREPARATION

## B.1    DATASET STATISTICS

We preprocess each dataset by extracting user–query–LLM–response tuples and partition them into train, validation, and test sets. To ensure fair evaluation and meaningful personalization, we stratify the splits to maintain balanced user–model preference distributions and avoid degenerate cases (e.g., users consistently preferring a single LLM or lacking query diversity). This setup promotes generalization under cold-start conditions and supports robust evaluation of routing behavior.

For ChatBot Arena, we selected the following users and LLMs:

**Users:** arena_user_9965, arena_user_15085, arena_user_257, arena_user_13046, arena_user_11473, arena_user_3820, arena_user_9676, arena_user_6467, arena_user_6585, arena_user_5203, arena_user_1338

**LLMs:** koala-13b, vicuna-13b, gpt-3.5-turbo, oasst-pythia-12b, gpt-4, claude-v1, RWKV-4-Raven-14B, palm-2, alpaca-13b, mpt-7b-chat, vicuna-7b, claude-instant-v1, chatglm-6b, fastchat-t5-3b, dolly-v2-12b, stablelm-tuned-alpha-7b

We report in Table 6 the size of each dataset along with the time required to process it into the heterogeneous graph used in our experiments.

Table 5: Dataset statistics, including the number of entries, users, and LLMs in each split.

| Dataset | Split | #Entries | #Users | #LLMs |
|---------|-------|----------|--------|-------|
| Chatbot-Arena | Train | 2780 | 11 | 16 |
| | Valid | 386 | 11 | 16 |
| | Test | 824 | 11 | 16 |
| MT-Bench | Train | 2240 | 10 | 2 |
| | Valid | 320 | 10 | 2 |
| | Test | 640 | 10 | 2 |
| GSM8K | Train | 18460 | 10 | 2 |
| | Valid | 2620 | 10 | 2 |
| | Test | 5300 | 10 | 2 |
| MMLU | Train | 3970 | 5 | 2 |
| | Valid | 560 | 5 | 2 |
| | Test | 1150 | 5 | 2 |

Table 6: Computational cost of graph construction across datasets.

| Dataset | Data Entries | Avg. Tokens | Encoding Time (s) | Graph Construction Time (s) |
|---------|--------------|-------------|-------------------|------------------------------|
| ChatBot-Arena | 3990 | 184.41 | 51.73 | 1.70 |
| MT-Bench | 3200 | 4511.73 | 55.68 | 2.40 |
| GSM8K | 26380 | 112.68 | 142.84 | 1.49 |
| MMLU | 5680 | 9.35 | 4.27 | 1.56 |
| LaMP | 9850 | 66.82 | 30.98 | 1.91 |

## B.2 SYNTHETIC USER DESIGN

To simulate diverse user preferences, we introduce synthetic users whose routing behavior is governed by a weighted linear utility function over multiple metrics: human preference rating, token count, output diversity, and cost. For each dataset, we manually assign different weights $\{w_{\text{rating}}, w_{\text{tokens}}, w_{\text{diff}}, w_{\text{cost}}\}$ per user to reflect individualized trade-offs, such as favoring cost-efficiency or output diversity over raw model quality. These weights are normalized within each dataset to prevent scale bias.

Table 7: **Synthetic user weights for MT-Bench dataset.**

| User | $w_{\text{rating}}$ | $w_{\text{tokens}}$ | $w_{\text{diff}}$ | $w_{\text{cost}}$ |
|------|---------|---------|--------|--------|
| user_1 | 1.42 | 0.0087 | −0.174 | −45.23 |
| user_2 | 1.87 | 0.0012 | 0.091 | −15.55 |
| user_3 | 0.96 | 0.0135 | 0.045 | −48.42 |
| user_4 | 1.15 | −0.0008 | −0.220 | −10.00 |
| user_5 | 1.69 | 0.0024 | 0.175 | −38.50 |
| user_6 | 1.08 | −0.0015 | −0.030 | −25.12 |
| user_7 | 0.53 | 0.0162 | 0.230 | −5.75 |
| user_8 | 1.34 | −0.0005 | −0.145 | −12.40 |
| user_9 | 1.98 | 0.0101 | 0.087 | −25.10 |
| user_10 | 1.57 | 0.0024 | −0.065 | −7.79 |

## B.3 PROCESSING OF THE LaMP DATASET

We select the "Personalized Scholarly Title Generation" task from the LaMP benchmark (Salemi et al., 2024b) as our new dataset. This task provides pairs of paper titles and abstracts for multiple users and requires predicting the title a user would prefer given an abstract.

Table 8: **Synthetic user weights for GSM8K dataset.**

| User | $w_{\text{rating}}$ | $w_{\text{tokens}}$ | $w_{\text{diff}}$ | $w_{\text{cost}}$ |
|---|---|---|---|---|
| user_1 | 1.0 | 20.0 | 100.0 | -0.0 |
| user_2 | 1.5 | 18.0 | 50.0 | -1.0 |
| user_3 | 0.8 | 22.0 | 80.0 | -0.5 |
| user_4 | 1.2 | 17.0 | 120.0 | -0.2 |
| user_5 | 2.0 | 15.0 | 70.0 | -0.4 |
| user_6 | 0.4 | 6.0 | -4.0 | -1.0 |
| user_7 | 0.3 | 7.0 | -5.0 | -0.9 |
| user_8 | 0.6 | 8.0 | -7.0 | -1.2 |
| user_9 | 0.2 | 9.0 | -9.0 | -0.8 |
| user_10 | 0.8 | 10.0 | -3.0 | -1.1 |

Table 9: **Synthetic user weights for MMLU dataset.**

| User | $w_{\text{rating}}$ | $w_{\text{tokens}}$ | $w_{\text{diff}}$ | $w_{\text{cost}}$ |
|---|---|---|---|---|
| user_1 | 1.0 | 0.00 | 0.00 | 0.0 |
| user_2 | 1.0 | 0.00 | 0.00 | $-600.0$ |
| user_3 | 1.0 | 0.00 | 0.00 | $-1200.0$ |
| user_4 | 1.0 | 0.00 | 0.00 | $-1800.0$ |
| user_5 | 1.0 | 0.00 | 0.00 | $-2400.0$ |

**Data extraction.** We identify the 10 users with the largest amount of data and randomly sample 200 (title, abstract) pairs for each user.

**LLM response generation.** We use five LLMs with diverse architectures and sizes—deepseek-r1 (DeepSeek-AI, 2024), gemma-2-27b-it (Team, 2024), llama-3.1-8b-instruct (AI@Meta, 2024), qwen2-7b-instruct (qwe, 2024), and mistral-7b-instruct-v0.3 (Jiang et al., 2023)—to generate a predicted title for each abstract.

**User rating acquisition.** For each paper, we encode both the ground-truth title and all LLM-generated titles using a PLM. We compute the cosine similarity between a generated title and the ground-truth title and treat this score as the user rating.

**Dataset filtering and splitting.** For each abstract, we identify the LLM with the highest user rating and use it as the routing target, discarding samples where ties occur. This yields a total of 9,850 instances, which we split into training, validation, and test sets using a 7:1:2 ratio.

## C   BASELINE ROUTING PROMPTS

To benchmark routing strategies, we design two representative prompt templates: one for a vanilla router that selects the best LLM without personalization, and another for a personalized router that incorporates user history and preferences. Both prompts simulate realistic routing scenarios where a system must choose a single LLM for the next turn in a multi-turn dialogue.

## D   ADDITIONAL RESULTS FOR CASE STUDIES

### D.1   GENERALIZATION

Here, we present the results of the experiments described in Section 5.2 on the ChatBot Arena and MT-Bench datasets, as shown in Figures 6 and 7 respectively.

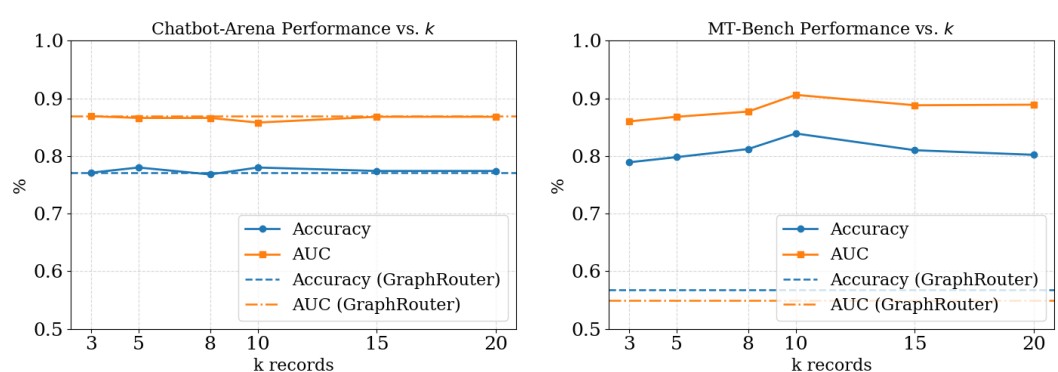

Figure 6: K-selection across datasets.

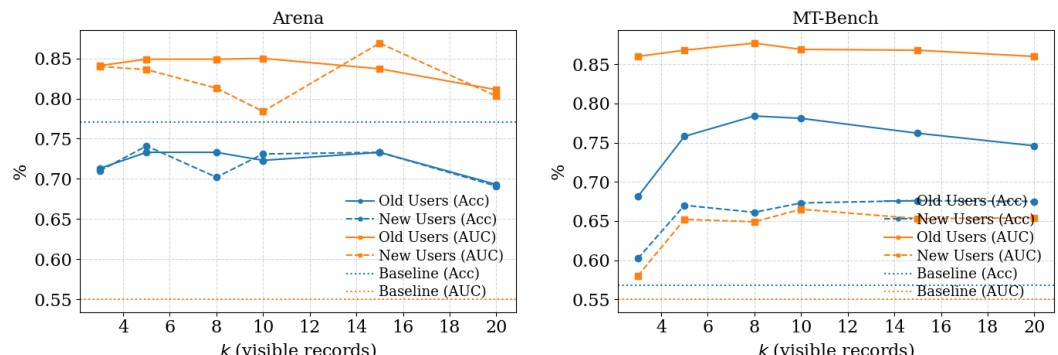

Figure 7: Generalization to new users.

Table 10: **Prompt Template: Vanilla LLM Routing (No Personalization)**

**[Instruction]**
You are an expert routing agent. Your task is to select the most suitable Large Language Model (LLM) to handle the next query in a multi-turn conversation.
**[Input Format]**
```
[Candidate LLM List]
{{CANDIDATE_LLM_LIST}}
[Previous Conversation]
{{PREVIOUS_CONVERSATION}}
[Current Query]
{{CURRENT_QUERY}}
```
**[Instructions for Model Selection]**

- Consider the query difficulty, the context of the previous conversation, and each LLM's expertise, cost, and size.
- Choose the single best LLM to respond to the current query.
- Output only the name of the selected LLM in the exact format below.
- Do not provide explanations or commentary.

**[Output Format]**
```
<'{selected_model_name}'>
```

Table 11: **Prompt Template: Personalized Routing (User History Aware)**

**[Instruction]**
You are an expert routing agent. Your task is to select the most suitable Large Language Model (LLM) to handle the next query in a multi-turn conversation, incorporating both model characteristics and personalization signals from the user's history.
**[Input Format]**
```
[Candidate LLM List]
{{CANDIDATE_LLM_LIST}}
[Previous Conversation]
{{PREVIOUS_CONVERSATION}}
[Current Query]
{{CURRENT_QUERY}}
[User Preference History]
{{USER_PREFERENCE_HISTORY}}
```
**[Instructions for Model Selection]**

- Consider the query difficulty, the context of the ongoing conversation, the LLMs' specializations, cost, and size.
- Additionally, factor in the user's historical preferences and ratings to personalize the routing decision.
- Choose the single best LLM to respond to the current query.
- Output only the name of the selected LLM in the exact format below.
- Do not provide explanations or commentary.

**[Output Format]**
```
<'{selected_model_name}'>
```

# E    THE USE OF LARGE LANGUAGE MODELS (LLMS)

During the writing of this paper, we used the GPT-5 Mini model for text polishing and grammatical corrections to enhance the readability of the manuscript.

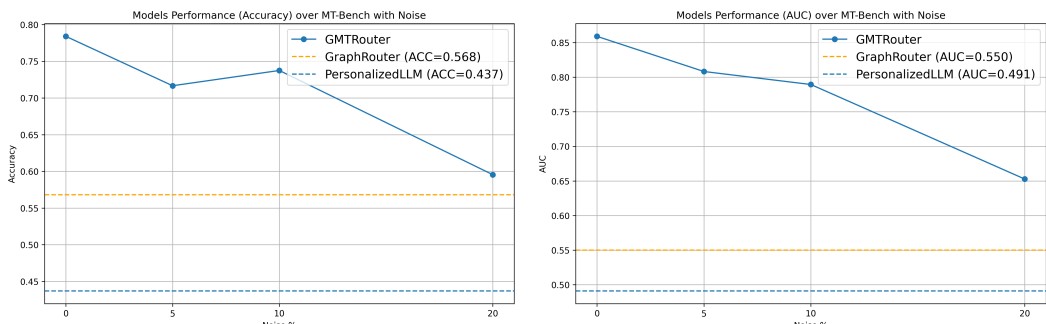

Figure 8: Performance against noisy preference over MT-Bench

# F  FURTHER EXPERIMENT

## F.1  EXTENDED EXPERIMENTS ON CHATBOT ARENA USERS

Table 12: GMTRouter performance on the ChatBot Arena dataset with newly added users.

|          | New Users = 0 | New Users = 3 | New Users = 161 |
|----------|---------------|---------------|-----------------|
| Accuracy | 0.774         | 0.780         | 0.790           |
| AUC-ROC  | 0.875         | 0.858         | 0.837           |

To more thoroughly validate that GMTRouter exhibits strong generalization to new users, we further selected all ChatBot Arena users with more than 15 historical interactions (161 users in total). We keep the original training and test sets used in Section 5 unchanged and use all newly added data exclusively as the test set. GMTRouter is then evaluated on these newly added users. The experimental results are shown in Table 12.

This expanded evaluation enables us to assess GMTRouter on a substantially larger user set and further demonstrates its effectiveness and generalization capability.

## F.2  EXPERIMENTS WITH NOISY DATA

We conduct experiments to evaluate GMTRouter's personalization performance under varying levels of noisy data. Specifically, we use the MT-Bench dataset and swap the user ratings of $k\%$ of the data ($k = 5, 10, 20$), simulating noise and inconsistencies in preference signals (Li et al., 2024a). The experimental results are shown in Figure 8. Our results indicate that, although performance naturally decreases as noise increases, GMTRouter **degrades gracefully and remains competitive even with 20% noise, outperforming the strongest baseline trained on clean data.** We attribute this robustness to **user-conditioned graph sampling**, which aggregates signals across multiple interactions and mitigates the impact of individual noisy labels.

## F.3  COMPARISON WITH PERSONALIZED GENERATION METHOD

To evaluate the personalization capabilities of GMTRouter against personalized generation approaches, we adopt the untuned In-Prompt Augmentation (IPA) based Retrieval Augmented Generation (RAG) from the LaMP benchmark (Salemi et al., 2024b), employing Contriever (Izacard et al., 2021) as the retrieval backbone.

Experiments were conducted on the LaMP dataset. For the RAG baseline, we utilized the test set's 'abstract' query to retrieve the **top-10** most relevant historical 'abstracts' from the specific user's training data. These retrieved items were formatted as 'title, abstract' pairs and integrated into the LLM's input as few-shot examples. Crucially, while GMTRouter relies on user ratings (calculated by comparing the LLM's predicted titles against the ground-truth titles) as its supervision signal,

we provided the RAG baseline with a **more direct and potent** form of supervision by explicitly incorporating the retrieved ground-truth titles into the few-shot examples.

We employ **average user rating** and **average token usage** as our primary evaluation metrics. The GMTRouter output consists of the raw response generated by the routed LLM, whereas the RAG output is generated by DeepSeek-R1 (DeepSeek-AI, 2024), conditioned on the prompt augmented with the retrieved few-shot examples. Furthermore, we also investigated the personalization capabilities of combining both GMTRouter and RAG. Specifically, GMTRouter is employed to select the optimal LLM, and the RAG is subsequently used to construct the few-shot augmented prompt. The experimental results are presented in Table 13.

Table 13: Performance Comparison of GMTRouter and IPA-RAG on the LaMP Benchmark. The **Random Routing** column serves as the **lower bound**, representing the expected user rating achieved by randomly selecting an LLM. Conversely, the **Theoretical Best Routing** column establishes the **upper bound** for the routing task, reflecting the user rating obtained by always selecting the LLM that yields the highest user rating for that specific instanc.

| Metric | Random Routing | GMTRouter | IPA-RAG | GMTRouter + IPA-RAG | Theoretical Best Routing |
|---|---|---|---|---|---|
| AVG User Rating | 0.744 | 0.772 | 0.775 | **0.784** | 0.810 |
| AVG Token Cost | – | 293.56 | 3231.72 | 2358.89 | – |

Our experiments lead to three key observations:

1. **Comparable Personalization:** Both GMTRouter (routing) and the personalized generation approach (IPA-RAG) achieve comparable improvements in personalization capability as measured by the AVG User Rating.

2. **Cost Disparity:** The IPA-RAG baseline incurs a significantly higher cost, requiring several times more tokens than GMTRouter, highlighting the efficiency gains offered by the routing mechanism.

3. **Synergistic Effect:** Combining the two methods (GMTRouter + IPA-RAG) yields the best empirical performance. This suggests that routing and personalized generation techniques address distinct, complementary facets of personalization.

Table 14: GMTRouter performance using different heterogeneous convolutional layers.

| Conv Layer | Chatbot-Arena | MT-Bench | GSM8K | MMLU |
|---|---|---|---|---|
| | ACC / AUC | ACC / AUC | ACC / AUC | ACC / AUC |
| HeteroConv | 0.777 / 0.867 | 0.569 / 0.492 | 0.499 / 0.603 | 0.494 / 0.542 |
| HANConv | 0.766 / 0.776 | 0.646 / 0.680 | 0.774 / 0.775 | 0.707 / 0.746 |
| HGTConv | 0.774 / 0.875 | 0.784 / 0.859 | 0.773 / 0.859 | 0.771 / 0.870 |

## F.4 USING DIFFERENT HETEROGENEOUS GNNS AS THE GMTROUTER BACKBONE

We investigate how different heterogeneous GNNs used as the GMTRouter backbone affect its personalization capability. We evaluate three backbones: HGT (Ziniu Hu, 2020), HAN Wang et al. (2019), and HeteroConv, and present the results in Table 14.

We observe that attention-based heterogeneous GNNs (HAN, HGT) consistently outperform the simpler aggregation-based HeteroConv backbone. These results also indicate that GMTRouter is not dependent on any particular GNN architecture: multiple attention-based backbones achieve strong performance, suggesting that the improvements mainly stem from the graph-based personalized routing framework and data modeling, rather than from a specific convolutional operator.

