# OpenReview forum: "GMTRouter: Personalized LLM Router over Multi-turn User Interactions"
_ICLR.cc/2026/Conference — ICLR 2026 Conference Withdrawn Submission_

### Official Review · Reviewer_8ann · 2025-10-29

**Soundness:** 2
**Presentation:** 3
**Contribution:** 2
**Rating:** 4
**Confidence:** 2

**Summary:**

This paper introduces GMTRouter, a personalized LLM router for LLMs that extracts structured preference information from multi-turn user-LLM interactions. The methodology is to model user-LLM interactions as a heterogeneous graph with different node types (users, queries, responses, LLMs, and virtual "turn" nodes) to capture the relational structure among entities. Then, a heterogeneous graph transformer (HGT) is used to predict which LLM best suits a given user-query pair. Experiments on Chatbot Arena, MT-Bench, GSM8K, and MMLU show consistent improvements over baselines such as GraphRouter and FrugalGPT. The authors also show that GMTRouter generalizes well to new users and remains computationally efficient.

**Strengths:**

1. Modeling multi-turn user-LLM interactions as a heterogeneous graph with explicit node types and virtual "turn" nodes is a novel approach to personalized routing.
2. The inductive training strategy allows adaptation to new users with minimal interaction data, addressing the common cold-start problem.
3. The framework is lightweight and practical, requiring modest computational resources, which enhances its potential for real-world deployment in LLM routing systems.

**Weaknesses:**

1. The design of GMTRouter is largely empirical and heuristic. There is no theoretical justification for why the proposed graph structure and the HGT-based model can better extract and generalize user preferences compared to other models.
2. The main novelty lies in the data modeling design (heterogeneous graph construction), while the learning components (e.g., HGT backbone, inductive training) largely follow existing graph learning literature.
3. There are many duplicated references, e.g., (Chen et al., 2023a) and (Chen et al., 2023b), (Chen et al., 2023a) and (Chen et al., 2023b), (Feng et al., 2024a) and (Feng et al., 2024b), (Hu et al., 2020a) and (Hu et al., 2020b), (Ong et al., 2024a) and (Ong et al., 2024b).

Minor issues:
1. The abbreviation "AUC" is not defined in the paper.

**Questions:**

1. As the heterogeneous graph is constructed directly from the interaction history, it should not contain more information than the history itself. Could the authors elaborate on why such a graph-based representation improves preference elicitation compared to directly learning from the raw interaction table?
2. The graph construction includes multiple users within the same graph (e.g., Figure 3(b)). However, user preferences are typically independent of one another. Would constructing individual user-specific graphs (and learning over them) be more intuitive? How does the shared multi-user graph influence personalization performance?

---

> ### Author Response · Authors · 2025-11-25
> **Author response to Reviewer 8ann (1/3)**
>
> ***W1.The design of GMTRouter is largely empirical and heuristic. There is no theoretical justification for why the proposed graph structure and the HGT-based model can better extract and generalize user preferences compared to other models.***
>
> **Author Response:**
>
> Thanks for raising the concern.
> The focus of GMTRouter is on a specific problem: **personalized routting task**. In this setting, we follow the standard practice in recent LLM and graph frameworks of prioritizing **rigorous empirical evidence** over formal theory. GMTRouter is evaluated on multiple datasets and against a broad suite of baselines, and we conduct ablations on each core component (graph structures, sampling strategy and backbone) to validate their individual contributions ***[Details in § 5]***. This empirical-first approach is consistent with many influential, framework-style works that build on existing components, such as Chain-of-Thought prompting [1], Self Consistency CoT [2], Toolformer [3], and SPECTER [4].
>
> Following the reviewer’s comment, we further experimented with additional GNN architectures as the backbone of GMTRouter. The results are summarized below:
>
> **Table: GMTRouter with different heterogeneous convolutional layers**
> | Conv Layer | Chatbot-Arena | MT-Bench | GSM8K | MMLU |
> |--------|:--------:|:--------:|:--------:|:--------:|
> |        | ACC / AUC | ACC / AUC | ACC / AUC | ACC / AUC ||
> | HeteroConv | 0.777 / 0.867 | 0.569 / 0.492 | 0.499 / 0.603 | 0.494 / 0.542 |
> | HANConv | 0.766 / 0.776 | 0.646 / 0.680 | 0.774 / 0.775 | 0.707 / 0.746 |
> | HGTConv | 0.774 / 0.875 | 0.784 / 0.859 | 0.773 / 0.859 | 0.771 / 0.870 |
>
> We observe that attention-based heterogeneous GNNs (HAN, HGT) consistently outperform the simple aggregation-based HeteroConv backbone. At the same time, the results show that GMTRouter is not tied to HGT specifically.
>
> We have added these experiments and discussions to the revised paper ***Details Appendix F.4***. We hope this clarifies that
> - Our design choices supported by systematic empirical evidence.
> - GMTRouter should be viewed as a **general framework**, which can be instantiated with different heterogeneous GNN backbones ***[Discussed in Line 238-241]***, rather than a claim of theoretical optimality for a specific architecture.
> - It serve as a pivot for future works to futher enhance personalized routing.
>
> A formal theoretical characterization of when such heterogeneous graph architectures provably outperform alternative representations is an interesting direction for future work, but is beyond the scope of this paper.
>
> **[1]** Wei, Jason, et al. "Chain-of-thought prompting elicits reasoning in large language models." Advances in neural information processing systems 35 (2022): 24824-24837.
>
> **[2]** Wang, Xuezhi, et al. "Self-consistency improves chain of thought reasoning in language models." arXiv preprint arXiv:2203.11171 (2022).
>
> **[3]** Schick, Timo, et al. "Toolformer: Language models can teach themselves to use tools." Advances in Neural Information Processing Systems 36 (2023): 68539-68551.
>
> **[4]** Cohan, Arman, et al. "Specter: Document-level representation learning using citation-informed transformers." arXiv preprint arXiv:2004.07180 (2020).

---

> ### Author Response · Authors · 2025-11-25
> **Author response to Reviewer 8ann (2/3)**
>
> ***W2.The main novelty lies in the data modeling design (heterogeneous graph construction), while the learning components (e.g., HGT backbone, inductive training) largely follow existing graph learning literature.***
>
> **Author Response:**
>
> We appreciate the reviewer’s comments regarding the novelty of our work. Here, we clarify the contributions and novel aspects of our research.
>
> GMTRouter is aimed to solve a previously unexplored problem **the personalized routing task *(§2.1)***. This task raises two concrete challenges:
>
> **(1) How to extract user preference signals from raw user-LLM interactions**:  As described in ***§3.1 and §3.2***, we propose a solution to **process the Interaction History Table into a heterogeneous graph, which maximally preserves the relational structure among key entities in the interactions**. This allows the model to aggregate multi-turn structure, propagate and encode preference information. We further design mechanisms to address inconsistencies in the format of preference data.
>
> **(2) How to make accurate predictions when user data is scarce**: Unlike approaches that rely on large amounts of data to update node representations, **GMTRouter is explicitly designed to capture a user’s preferences from only a few interaction records**. Concretely, we employ **user-conditioned graph sampling** during training, where each iteration selects only a small subset of data for message passing (10 samples per user) ***[Details disclosed in Line 247-261]***. This restriction on the amount of data involved in message passing **encourages the model to learn how to infer user preferences from very limited signals and to generalize efficiently to new users**. Our **Prediction Head** then maps the aggregated representations to preference scores, completing the **lightweight end-to-end routing pipeline**. ***[Details are described in Line 262-277]***
>
> To our knowledge, **this is the first work to formalize and tackle the personalized routing task**. Our main contribution is the design of the heterogeneous graph framework and training scheme for few-shot personalization, rather than a new GNN operator.
>
> ***W3. There are many duplicated references***
>
> **Author Response:**
>
> We thank the reviewer for pointing out the issue of duplicated references.
> We have corrected this in the updated version, and, once again, we appreciate the reviewer’s feedback for helping us make the paper more professional and well-formatted.
>
> ***Issue: The abbreviation "AUC" is not defined in the paper.***
>
> **Author Response:**
>
> We thank the reviewer for pointing out the missing definition in our paper. "AUC" is defined as Area under the Receiver Operating Characteristic Curve (AUC-ROC) using the following formula:
> $$\mathrm{AUC} = \Pr(s_{+} > s_{-}) + \frac{1}{2}\Pr(s_{+} = s_{-})$$
> Where $s_{+}$ and $s_{-}$ denote the scores of positive and negative instances respectively.
>
> We have added it in the updated version at ***[Line 353-358]***.

---

> ### Author Response · Authors · 2025-11-25
> **Author response to Reviewer 8ann (3/3)**
>
> ***Q1. As the heterogeneous graph is constructed directly from the interaction history, it should not contain more information than the history itself. Could the authors elaborate on why such a graph-based representation improves preference elicitation compared to directly learning from the raw interaction table?***
>
> **Author Response:**
>
> We thank the reviewer for raising the question regarding the advantages of a graph-based representation. While the topological structure of the graph is indeed derived from the interaction history, **our graph-based approach provides superior preference elicitation because it explicitly models the relationships among key entities in the interactions.**
>
> GMTRouter extracts the key entities from the interaction history and carefully designs the graph structure to capture the rich relational patterns within the interactions. Through message passing, each node explicitly aggregates information from its related nodes, dynamically refining its representation. **The graph structure imposes a strong relational inductive bias, allowing the model to capture user preferences more effectively than treating interaction rows independently.**
>
> To further ground this discussion, **we selected TIGER [5], a recent method that learns directly from raw interaction data, and RouteLLM [6] as new baselines**. The experimental results are shown below:
>
> **Table of TIGER & GMTRouter performance over datasets**
>
> | Model | Chatbot-Arena | MT-Bench | GSM8K | MMLU | LaMP |
> |--------|:--------:|:--------:|:--------:|:--------:|:--------:|
> |        | ACC / AUC | ACC / AUC | ACC / AUC | ACC / AUC | ACC / AUC |
> | RouteLLM   | 0.492 / 0.485 | 0.480 / 0.475 | 0.499 / 0.498 | 0.532 / 0.513 | - |
> | TIGER  | 0.739 / 0.735 | 0.656 / 0.691 | 0.639 / 0.683 | 0.710 / 0.764 | 0.339 / 0.698 |
> | GMTRouter  | 0.774 / 0.875 | 0.784 / 0.859 | 0.773 / 0.859 | 0.771 / 0.870 | 0.349 / 0.662 |
>
> *Note that RouteLLM is designed for routing between two LLMs, whereas the LaMP dataset requires routing among five LLMs; therefore, RouteLLM was not evaluated on LaMP.*
>
> Moreover, as reported in ***Table 2***, we have included several non-graph-based baselines (e.g FrugalGPT, Personalized LLM) as comparisons.
> The results confirm that representing the interaction history as a heterogeneous graph enables GMTRouter to achieve more effective personalization compared to learning directly from raw data. These empirical results complement the conceptual argument: the graph structure imposes a relational inductive bias that  encourages the model to exploit dependencies across users, queries, responses, and LLMs, rather than treating interaction rows as independent.
>
> ***Q2. The graph construction includes multiple users within the same graph (e.g., Figure 3(b)). However, user preferences are typically independent of one another. Would constructing individual user-specific graphs (and learning over them) be more intuitive? How does the shared multi-user graph influence personalization performance?***
>
> **Author Response:**
>
> We thank the reviewer for this insightful question. As the reviewer correctly notes, user preferences are diverse, and signals from different users can interfere with one another.
> Although we conceptually construct a global interaction graph (Fig. 3(b)), **message passing is always performed on per-user subgraphs**:
> - During training ***[Line 248-251]***, each sampled subgraph only contains interactions from a single user.
> - During inference ***[Line 273-277]***, we also build a subgraph using only that user's history and candidate LLMs.
>
> Thus, in practice, GMTRouter operates on **individual user-specific graphs**, and there is no cross-user message passing; the global graph is simply the union of these user-specific components for convenience of storage and indexing. Personalization arises from user-specific node representations together with parameter sharing across users.
> We have clarified these details in the paper and will update ***Figure 3*** in future revisions to better illustrate the design. We, once again, thank the reviewer for the insightful comments to help us better polish our work.
>
> **[5]** Rajput, Shashank, et al. "Recommender systems with generative retrieval." Advances in Neural Information Processing Systems 36 (2023): 10299-10315.
>
> **[6]** Ong, Isaac, et al. "Routellm: Learning to route llms with preference data." arXiv preprint arXiv:2406.18665 (2024).

---

### Official Review · Reviewer_L1od · 2025-10-30

**Soundness:** 2
**Presentation:** 3
**Contribution:** 2
**Rating:** 4
**Confidence:** 4

**Summary:**

The paper proposes GMTRouter, a personalized LLM routing framework that models multi-turn user–LLM interactions as a heterogeneous graph. By utilizing a lightweight inductive graph learning framework, GMTRouter captures user preferences from few-shot data and adapts to new users.

**Strengths:**

1. The proposed GMTRouter models multi-turn user–LLM interactions using a heterogeneous graph, capturing complex relational dependencies for personalization.
2. The inductive graph learning framework allows GMTRouter to adapt to new users with minimal data.
3. Experimental results show that GMTRouter consistently outperforms baselines across multiple datasets.

**Weaknesses:**

1. The paper heavily relies on LLM routing for personalization, but this essentially requires the model itself to have inherent personalization capabilities, making the routing process somewhat secondary.
2. Memory construction is a promising approach for personalization, but the paper does not provide a comparison with existing memory-based methods.
3. The graph construction process is time-intensive; however, the paper lacks a detailed analysis of the cost involved in updating the graph post-construction.

**Questions:**

The related work section does not sufficiently cover the wide range of recent LLM personalization efforts. More attention should be given to the existing literature on personalized LLMs.
The paper does not provide a comparison with existing personalized LLM methods.

---

> ### Author Response · Authors · 2025-11-25
> **Author response to Reviewer L1od (1/2)**
>
> ***W1.The paper heavily relies on LLM routing for personalization, but this essentially requires the model itself to have inherent personalization capabilities, making the routing process somewhat secondary.***
>
> **Author Response:**
>
> We thank the reviewer for raising this important conceptual point.
> While **personalized generation** is indeed a powerful direction, **personalized routing addresses a complementary and equally important problem.**
> - Personalized routing **allows the system to choose the most suitable LLM for user-query, balancing the answer quality, computational cost, response style, and other factors based on a user’s individual preferences.**
> - Routing does not require the LLM itself to be inherently personalized. Instead, it **maps a user’s history to the model whose native strengths best align with that user.**
>
> Thus, personalization and routing are orthogonal and can be combined to further enhance user-LLM interactions.
>
> Following the suggestion, we added IPA-RAG from LaMP [1] as a representative personalized-generation method and evaluated it on our newly added LaMP dataset ***[Details: Line 296-300]***, comparing its personalization performance gains with GMTRouter. In addition, we further evaluated the gains achieved when combining GMTRouter and IPA-RAG. The results are reported below and in the revised paper. ***[Details: Appendix F.3 and Table 13]***
>
> **Table of GMTRouter and IPA-RAG performance over LAMP**
> |           | Random routing | GMTRouter | IPA-RAG  | GMTRouter + IPA-RAG | Theoretical Best Routing|
> | :---:     | :---: | :---: | :---: | :---: | :---: |
> | AVG User Rating  | 0.744 | 0.772 | 0.775  | 0.784 | 0.810 |
> | AVG Token Cost   | - | 293.56 | 3231.72 | 2358.89 | - |
>
> Our experiments show that
> - **Comparable Personalization:** Both GMTRouter (routing) and the personalized generation approach (IPA-RAG) achieve comparable improvements in personalization capability as measured by the AVG User Rating.
> - **Cost Disparity:** The IPA-RAG baseline incurs a significantly higher cost, requiring several times more tokens than GMTRouter, highlighting the efficiency gains offered by the routing mechanism.
> - **Synergistic Effect:** Combining the two methods (GMTRouter + IPA-RAG) yields the best empirical performance. This suggests that routing and personalized generation techniques address distinct, complementary facets of personalization.
>
> ***W2. Memory construction is a promising approach for personalization, but the paper does not provide a comparison with existing memory-based methods.***
>
> **Author Response:**
>
> We appreciate the reviewer’s suggestion to include a memory-based personalization method for comparison. Following this suggestion, **we have added MA-GNN [2], a memory-based graph model, as a new baseline *[Detailed explaination in Line 338-342]***, and the results are shown below:
>
> **Table of MA-GNN & GMTRouter performance across datasets**
> | Model     | Chatbot-Arena | MT-Bench | GSM8K | MMLU | LaMP |
> |------------|:-------------:|:--------:|:-----:|:----:|:----:|
> |            | ACC / AUC     | ACC / AUC | ACC / AUC | ACC / AUC | ACC / AUC |
> | MA-GNN   | 0.673 / 0.775 | 0.679 / 0.739 | 0.636 / 0.648 | 0.702 / 0.758 | 0.347 / 0.661 |
> | GMTRouter  | 0.774 / 0.875 | 0.784 / 0.859 | 0.773 / 0.859 | 0.771 / 0.870 | 0.349 / 0.662 |
>
> These results shows that **GMTRouter consistenly outperforms the memory-based method** , validating its ability to capture preference information more effectively.  The new results have been incorporated into ***Table 2***.
>
> **[1]** Salemi, Alireza, et al. "Lamp: When large language models meet personalization." Proceedings of the 62nd Annual Meeting of the Association for Computational Linguistics (Volume 1: Long Papers). 2024.
>
> **[2]** Ma, Chen, et al. "Memory augmented graph neural networks for sequential recommendation." Proceedings of the AAAI conference on artificial intelligence. Vol. 34. No. 04. 2020.

---

> ### Author Response · Authors · 2025-11-25
> **Author response to Reviewer L1od (2/2)**
>
> ***W3.The graph construction process is time-intensive; however, the paper lacks a detailed analysis of the cost involved in updating the graph post-construction.***
>
> **Author Response:**
>
> We thank the reviewer for pointing this out.
>
> GMTRouter’s primary time cost during graph construction lies in generating initial embeddings for each node using a pretrained language model. To keep the process lightweight, we adopt Contriever [3], a compact BERT-based model, as our PLM. Following the reviewer’s suggestion, **we now include the sizes of all datasets used and the corresponding time required to construct their graphs.** Details are reported below and updated in ***[Appendix B.1 and Table 6]***
>
> **Table of computational cost of graph construction**
> |           | # data entry | Average tokens per entry | Encoding Time (s) | Graph Constructing Time (s) |
> | :---:     | :---: | :---: | :---: | :---: |
> | ChatBot-Arena | 3990 | 184.41     | 51.73 | 1.70 |
> | MT-Bench      | 3200 | 4511.73    | 55.68 | 2.40 |
> | GSM8K         | 26380 | 112.68    | 142.84| 1.49 |
> | MMLU          | 5682 | 9.35       | 4.27  | 1.56 |
> | LaMP          | 9850 | 66.82      | 30.98 | 1.91 |
>
>
> Once the initial embeddings are obtained, **both training and inference operate on small sampled subgraphs** (10 interaction histories per user), making the process highly efficient. For updating the graph after construction, **adding new user histories simply requires inserting nodes with their initial embeddings into the sampling pool, incurring only linear-time overhead with respect to the new data** and ensuring the scalability of the method.
>
>
> ***Q1.The related work section does not sufficiently cover the wide range of recent LLM personalization efforts. More attention should be given to the existing literature on personalized LLMs. The paper does not provide a comparison with existing personalized LLM methods.***
>
> **Author Response:**
>
> We appreciate the reviewer’s suggestion to add the discussion of personalized LLMs in the related work section. Following the reviewer’s suggestion, **we have added this discussion in the revised version of the paper *[Details: Line 516-522]***. In addition, as mentioned in our response to W1, **we have included comparisons with IPA-RAG, a personalized LLM method, and incorporated LaMP, a dataset specifically designed for LLM personalization, as an additional dataset**. We believe that evaluating GMTRouter against a broader range of methods and across more diverse domains further highlights its generality and efficiency in personalized routing.
>
> **[3]** Izacard, Gautier, et al. "Unsupervised dense information retrieval with contrastive learning." arXiv preprint arXiv:2112.09118 (2021).

---

### Official Review · Reviewer_s59p · 2025-10-31

**Soundness:** 4
**Presentation:** 2
**Contribution:** 2
**Rating:** 2
**Confidence:** 4

**Summary:**

GMTRouter tackles personalized LLM routing by modeling multi-turn user–LLM interactions as a heterogeneous graph whose nodes are users, LLMs, queries, and responses, with user feedback embedded as a preference feature. A lightweight inductive Heterogeneous Graph Transformer aggregates information over sampled conversation “turn” nodes, and a cross-attention prediction head ranks candidate LLMs for a new (user, query) pair using only a few past interactions. GMTRouter consistently outperforms baselines and generalizes well to unseen users with few-shot data. The model is compact and trainable on a single GPU.

**Strengths:**

1. The paper tests on one real-world and three synthetic benchmarks, builds multi-user labels that combine quality, cost, length, and rare-word signals, reports clear metrics, and assess generalization.
2. Practical, resource-efficient design suitable for deployment. GMTRouter is small, with modest storage and max GPU usage (~4.3 GB), and the experiments run on a single RTX A6000.

**Weaknesses:**

1. Missing baselines. It seems that the paper is missing some baselines achieving the personalization through graph. For example, Knowledge Graph Tuning: Real-time Large Language Model Personalization based on Human Feedback.
2. This paper is claiming it is a LLM routing based model, however, it does not compare to baselines of routed LLMs. For example, RouteLLM: Learning to Route LLMs with Preference Data.
3. What is the key challenge that this paper is trying to solve, all the modules seems pretty standard to the reviewer to form a GNN.
4. The writing is a little bit confusing. All previous parts are emphasizing the routing and selection, but the majority of the methods is on constructing the GNN, until the very last of the section, the reviewer can see how you really make the selection.

**Questions:**

See weaknesses.

---

> ### Author Response · Authors · 2025-11-25
> **Author response to Reviewer s59p (1/2)**
>
> ***W1. Missing baselines. It seems that the paper is missing some baselines achieving the personalization through graph. For example, Knowledge Graph Tuning: Real-time Large Language Model Personalization based on Human Feedback.***
>
> **Author Response:**
>
> We thank the reviewer for this helpful suggestion to include a graph-based personalization baseline. We have added a discussion of KGT in ***§ 6 Line 517-519***. However, since KGT is not open-sourced and its paper omits some implementation details, **we, instead, adopt another graph-based personalization method, MAGNN [1], as our new baseline *[Detailed explaination in Line 338-342]***. Results are shown in the table below and have been integrated into ***Table 2*** in the paper accordingly.
>
> ***W2. This paper is claiming it is a LLM routing based model, however, it does not compare to baselines of routed LLMs. For example, RouteLLM: Learning to Route LLMs with Preference Data.***
>
> **Author Response:**
>
> Thanks for this constructive suggestion to include a routed LLM baseline. We would first like to clarify that our method is **a graph-based framework for LLM routing**, rather than an LLM-routing-based model. Following the reviewer’s suggestion, **we added RouteLLM[2] as a new baseline and conducted the corresponding experiments *[Detailed explaination in Line 335-337]***.
>
> The new results are presented in the table below and have been incorporated into ***Table 2*** of the paper.
>
> **Table of RouteLLM & MA-GNN & GMTRouter performance across datasets**
> | Model     | Chatbot-Arena | MT-Bench | GSM8K | MMLU | LaMP |
> |------------|:-------------:|:--------:|:-----:|:----:|:----:|
> |            | ACC / AUC     | ACC / AUC | ACC / AUC | ACC / AUC | ACC / AUC |
> | RouteLLM   | 0.492 / 0.485 | 0.480 / 0.475 | 0.499 / 0.498 | 0.532 / 0.513 | - |
> | MA-GNN   | 0.673 / 0.775 | 0.679 / 0.739 | 0.636 / 0.648 | 0.702 / 0.758 | 0.347 / 0.661 |
> | GMTRouter  | 0.774 / 0.875 | 0.784 / 0.859 | 0.773 / 0.859 | 0.771 / 0.870 | 0.349 / 0.662 |
>
> *Note that RouteLLM is designed for routing between two LLMs, whereas the LaMP dataset requires routing among five LLMs; therefore, RouteLLM was not evaluated on LaMP.*
>
> These additions provide a more comprehensive comparison and demonstrate GMTRouter’s effectiveness across routing paradigms.
>
> **[1]** Ma, Chen, et al. "Memory augmented graph neural networks for sequential recommendation." Proceedings of the AAAI conference on artificial intelligence. Vol. 34. No. 04. 2020.
>
> **[2]** Ong, Isaac, et al. "Routellm: Learning to route llms with preference data." arXiv preprint arXiv:2406.18665 (2024).

---

> ### Author Response · Authors · 2025-11-25
> **Author response to Reviewer s59p (2/2)**
>
> ***W3. What is the key challenge that this paper is trying to solve, all the modules seems pretty standard to the reviewer to form a GNN.***
>
> **Author Response:**
>
> We thank the reviewer for raising this important point. The core contribution of our work is **not** introducing a new GNN layer, but instead, addressing a previously unstudied problem: **the personalized routing task (§2.1), which presents two fundamental challenges**:
>
> **(1) How to extract user preference signals from raw user-LLM interactions**:  As described in ***§3.1 and §3.2***, we propose a solution to **process the Interaction History Table into a heterogeneous graph, which maximally preserves the relational structure among key entities in the interactions**. This allows the model to aggregate multi-turn structure, propagate and encode preference information. We further design mechanisms to address inconsistencies in the format of preference data.
>
> **(2) How to make accurate predictions when user data is scarce**: Unlike approaches that rely on large amounts of data to update node representations, **GMTRouter is explicitly designed to capture a user’s preferences from only a few interaction records**. Concretely, we employ **user-conditioned graph sampling** during training, where each iteration selects only a small subset of data for message passing (10 samples per user) ***[Details disclosed in Line 247-261]***. This restriction on the amount of data involved in message passing **encourages the model to learn how to infer user preferences from very limited signals and to generalize efficiently to new users**.
> Additionally, our **Prediction Head** maps the aggregated representations to preference scores, completing the **lightweight end-to-end routing pipeline**. ***[Details are described in Line 262-277]***
>
> Thus, while the GNN components are standard, the problem we are targeting remains unexplored where naive full-graph training or non-personalized routers struggle.
>
> ***W4. The writing is a little bit confusing. All previous parts are emphasizing the routing and selection, but the majority of the methods is on constructing the GNN, until the very last of the section, the reviewer can see how you really make the selection.***
>
> **Author Response:**
>
> We greatly appreciate the reviewer for pointing out this potential confusion. We have revised ***§ 3*** to improved clarity.
> The goal of GMTRouter is to **make preference-aligned LLM selections**, and GNN components serve this goal by producing high-quality user representations. The structure is as follows:
> - ***§ 3.1-3.2***: Describe how interaction histories are transformed into a heterogeneous graph that captures the relational structure required for preference extraction.
> - ***§ 3.3***: Focus on **how user-conditioned subgraph sampling and message passing extract user embeddings**, and then makes explicit **how the Prediction Head converts these embeddings into LLM-selection scores.**
>
> GMTRouter is a **single, integrated pipeline**, where graph construction and aggregation components are necessary to represent user preference information, which are then used for routing decisions.

---

### Official Review · Reviewer_qr28 · 2025-11-01

**Soundness:** 3
**Presentation:** 3
**Contribution:** 2
**Rating:** 6
**Confidence:** 4

**Summary:**

This paper aims to tackle the task of personalized LLM routing: aligning model selection with individual user preferences based on their multi-turn interaction histories (to select the best LLM for the new query), which differs from existing works that are not personalized and often ignore multi-turn conversations. To tackle this, the authors proposed the GMTRouter, which represents users, queries, responses, and LLMs as nodes in a heterogeneous graph, allowing it to capture the relational structure across them. After that, through the message passing over the constructed graph, it enables predicting the suitable LLM for the new query. The authors validate the proposed approach on one realistic and three synthetic datasets, showing that it outperforms strong router baselines.

**Strengths:**

* The construction of the heterogeneous graph with four types of nodes (users, queries, responses, and LLMs) for personalized LLM routing is convincing.
* The proposed approach outperforms existing router baselines, even under the challenging scenarios (e.g., handling new users or with few data samples).
* The proposed router design is very efficient, requiring only minimal computing.

**Weaknesses:**

* The way that the authors construct the synthetic datasets for benchmarking and their quality is questionable. The datasets, such as MT-Bench, GSM8K, and MMLU, are not designed for personalization, and some of them are also not for multi-turn scenarios. How do the authors convert them to the multi-turn personalization settings? Additionally, how do you define users in those settings? More clarifications on them are needed.
* On the other hand, the scale of the realistic dataset (Chatbot Arena) is small, consisting of only 11 users. It would be great if the authors could scale this up.
* The novelty of the proposed method over the existing GraphRouter seems marginal, as the GraphRouter similarly constructs the heterogeneous graph and propagates messages between nodes to predict the LLM. It would be great if the authors could clarify whether the proposed approach is the extension of the GraphRouter, additionally considering the multi-turn interactions and users for personalization, or if there are any other aspects that the authors would like to emphasize.
* While the authors explicitly mention that the preference data by the single user is scarce, noisy, and inconsistent in format, it is questionable whether the authors validate the proposed approach on those challenging settings, and more importantly, whether the proposed approach has a certain design choice to tackle those challenges.
* It seems the authors reduce the margin for figures and tables at the top a lot, which should be fixed.

**Questions:**

Please see Weaknesses above.

---

> ### Author Response · Authors · 2025-11-25
> **Author response to Reviewer qr28 (1/2)**
>
> ***W1. The way that the authors construct the synthetic datasets for benchmarking and their quality is questionable. The datasets, such as MT-Bench, GSM8K, and MMLU, are not designed for personalization, and some of them are also not for multi-turn scenarios. How do the authors convert them to the multi-turn personalization settings? Additionally, how do you define users in those settings? More clarifications on them are needed.***
>
> **Author Response:**
>
> We appreciate the reviewer for raising questions regarding the construction of our synthetic datasets.
> Briefly, **we assign four scalar scores to the responses of multiple LLMs based on four distinct dimensions: Quality, Cost, Response Length, and Rare Words** ***[Mentioned in Line 306 - 311]***. Following the setting proposed in [1] and [2], we assume that different simulated users possess varying preferences across these four dimensions. These preferences are reflected by four distinct scalar coefficients ***[As discussed in Line 312 - 315 and Appendix B.2]***. The final score assigned by each user to a given response is then calculated as the inner product of the response score vector and the user coefficient vector.
>
> Regarding the concern over non-multi-turn settings, we affirm the point. To clarify, our proposed GMTRouter framework **is compatible with both single-turn and multi-turn settings**. Our experiments validate that GMTRouter achieves superior performance across both multi-turn datasets (MT-Bench, ChatBot Arena) and ones are single-turn.
>
> Furthermore, following the reviewer’s suggestion, **we have  incorporated one additional dataset specifically designed for personalization: LaMP[3].** We selected the “Personalized Scholarly Title Generation” task from LaMP and adapted it into a routing task dataset ***[Appendix B.3]***. We compared our method against various baselines, and the results are reported below:
>
> **Table of GMTRouter and baselines performances over LaMP dataset**
> |           | GMTRouter | GraphRouter   | PersonalizedLLM   | MA-GNN[4]| TIGER[5] |
> | :---:     | :---:     | :---:         | :---:             | :---: | :---: |
> | Accuracy  | **0.349** | $\underline{0.345}$  | 0.312             | 0.347 | 0.339 |
> | AUC-ROC   | $\underline{0.662}$     | 0.652  | 0.605             | 0.661 | **0.698** |
>
> We have updated the paper accordingly, and we believe this new evidence further highlights the efficiency of GMTRouter across diverse scenarios.
>
> ***W2. On the other hand, the scale of the realistic dataset (Chatbot Arena) is small, consisting of only 11 users. It would be great if the authors could scale this up.***
>
> **Author Response:**
>
> Thanks for the concerns regarding the scale of our experiments. In the ChatBot Arena setting, we initially selected the 11 users with the largest amount of data, as they **are representative and provide sufficient samples for training**. However, as stated in our paper, **GMTRouter exhibits strong generalization to new users**. Following the reviewer’s suggestion, we further **selected all ChatBot Arena users with more than 15 historical interactions (161 users in total)**. We keep the original training and test sets unchanged and use all newly added data exclusively as the test set. GMTRouter is then evaluated on these newly added users. The experimental results are shown below:
>
> **Table of GMTRouter performance over Chatbot-Arena dataset with new users**
> |           | New user = 0  | New user = 3  | New user = 161|
> | :---:     | :---:         | :---:         | :---:         |
> | Accuracy  | 0.774         | 0.780         | 0.790         |
> | AUC-ROC   | 0.875         | 0.858         | 0.837         |
>
>
> This expanded evaluation allows us to **assess GMTRouter on a significantly larger user set and further demonstrates its effectiveness and generalization ability**. We have incorporated this additional experiment into the revised paper. ***[Reported in Appendix F.1 and Table 12]***
>
>
>
> **[1]** Feng, Tao, Yanzhen Shen, and Jiaxuan You. "Graphrouter: A graph-based router for llm selections." arXiv preprint arXiv:2410.03834 (2024).
>
> **[2]** Feng, Tao, et al. "FusionFactory: Fusing LLM Capabilities with Multi-LLM Log Data." arXiv preprint arXiv:2507.10540 (2025).
>
> **[3]** Salemi, Alireza, et al. "Lamp: When large language models meet personalization." Proceedings of the 62nd Annual Meeting of the Association for Computational Linguistics (Volume 1: Long Papers). 2024.
>
> **[4]** Ma, Chen, et al. "Memory augmented graph neural networks for sequential recommendation." Proceedings of the AAAI conference on artificial intelligence. Vol. 34. No. 04. 2020.
>
> **[5]** Rajput, Shashank, et al. "Recommender systems with generative retrieval." Advances in Neural Information Processing Systems 36 (2023): 10299-10315.

---

> ### Author Response · Authors · 2025-11-25
> **Author response to Reviewer qr28 (2/2)**
>
> ***W3. The novelty of the proposed method over the existing GraphRouter seems marginal, as the GraphRouter similarly constructs the heterogeneous graph and propagates messages between nodes to predict the LLM. It would be great if the authors could clarify whether the proposed approach is the extension of the GraphRouter, additionally considering the multi-turn interactions and users for personalization, or if there are any other aspects that the authors would like to emphasize.***
>
> **Author Response:**
>
> Thanks for the evaluation over the novelty of our work. As the reviewer mentioned, GMTRouter is specifically designed for the multi-turn personalization scenario, introducing user nodes, response nodes (to transmit user preference signals), and turn nodes (to aggregate information from a single interaction turn), along with a more complex graph structure to facilitate information propagation. ***[Details: § 3.1 and § 3.2]***
>
> Moreover, unlike GraphRouter, which uses almost all nodes in the graph for message passing during training (see **Algorithm 1** in the GraphRouter paper for details), **GMTRouter is explicitly designed to capture a user’s preferences from only a few interaction records.** Concretely, we employ a **user-conditioned subgraph sampling** technique during training, where each iteration selects only a small subset of data for message passing (10 samples per user) ***[Details described in Line 247-261]***. This restriction on the amount of data involved in message passing **encourages the model to learn how to infer user preferences from very limited signals and to generalize efficiently to new users.** In addition, our **Prediction Head** maps the aggregated representations to preference scores, completing the **lightweight end-to-end routing pipeline**. ***[Details are described in Line 262-277]***.
>
> ***W4. While the authors explicitly mention that the preference data by the single user is scarce, noisy, and inconsistent in format, it is questionable whether the authors validate the proposed approach on those challenging settings, and more importantly, whether the proposed approach has a certain design choice to tackle those challenges.***
>
> **Author Response:**
>
> We sincerely thank the reviewer for highlighting this important concern.
> **Regarding scarcity:**  GMTRouter is explicitly designed to operate in data-scarce settings. During training, we apply an **inductive user-conditioned subgraph-sampling framework**, where each iteration samples only a small number of past interactions for each user. This forces the model to infer preferences from limited historical data. As shown in ***Figure 5 and § 5.2***, GMTRouter accurately captures a new user’s preferences using only a small amount of data (8–10 historical interactions).
>
> **Regarding noise:** Following the constructive suggestion, **we conducted an additional experiment in which we swapped the user ratings of k% of the data (k=5,10,20), simulating noise and inconsistency preference signals [6].** Results are shown below and reported in ***Appendix F.2 and Figure 8***. Although performance naturally decreases with more noise, **GMTRouter degrades gracefully and remains competitive even at 20% noise, outperforming clean-data baselines**. We attribute this robustness to **user-conditioned graph sampling**, which aggregates signals across multiple interactions and mitigates the impact of individual noisy labels.
>
> **Table: GMTRouter performance compare to baselines under noisy user preferences at MT-Bench dataset**
>
> |  | GMTRouter | GMTRouter | GMTRouter | GMTRouter | Personalized LLM | GraphRouter
> |-----------|:---------:|:---------:|:---------:|:---------:|:---------:|:---------:|
> | Noise Level |  0% | 5% | 10% | 20% | 0% | 0%|
> | ACC   | 0.784 | 0.717 | 0.738 | 0.638 | 0.437 | 0.568 |
> | AUC   | 0.859 | 0.808 | 0.789 | 0.653 | 0.491 | 0.550 |
>
> **Regarding inconsistencies in feedback format**: As described in ***Line 103-106***, we consider multiple forms of user feedback, including ratings, rankings, and ground-truth responses, and uniformly convert them into numerical ratings. This enables the router to learn consistently across datasets with different annotation styles ***[Details: Line 207-213]***. In our experiments, ChatBot Arena provides rankings as feedback ***[Line 302-303]***, MT-Bench, MMLU, and GSM8K use ratings ***[Line 306-315]***, and LaMP supplies ground-truth responses ***[Line 1004-1007]***.
>
> ***W5. It seems the authors reduce the margin for figures and tables at the top a lot, which should be fixed.***
>
> **Author Response:**
>
> We appreciate the reviewer for pointing out the formatting issues. We have corrected them and adjusted to the proper format in the new version.
>
> **[6]** Li, Haoxuan, et al. "Debiased recommendation with noisy feedback." Proceedings of the 30th ACM SIGKDD Conference on Knowledge Discovery and Data Mining. 2024.

---

### Author Response · Authors · 2025-12-04
**General Response by Authors**

We appreciate the reviewers’ efforts in evaluating our paper. Below, we summarize the key points reviewers raised—items marked with \*\* indicate issues for which we provide additional experiments or clarifications, while unmarked items reflect strengths acknowledged by them. "Action/Summary" includes the highly summarized rebuttal content for each reviewer.

| Dimension | Reviewer qr28 | Reviewer s59p | Reviewer L1od | Reviewer 8ann | Action/Summary |
| :--- | :--- | :--- | :--- | :--- | :--- |
| **Novelty & Design** | "Heterogeneous graph construction... is convincing" **"Novelty over GraphRouter marginal.."** | "Practical, resource-efficient design" **"Key challenge unclear.. modules standard"** | "Captures complex relational dependencies" **"Personalization vs. Routing conceptual.."** | "Explicit node types... is a novel approach" **"Design is empirical.. lacks theoretical justification"** | **Summary:** Reviewers generally find the graph design novel and practical. **Reviewer qr28 & s59p rebuttal:** We clarify the unique challenge of extracting preferences from scarce/noisy history and how GMTRouter differs from GraphRouter. **Reviewer 8ann rebuttal:** We added experiments to verify GMTRouter is a general framework effective across scenarios, rather than being designed for a specific GNN algorithm. |
| **Evaluation Scope** | **"Synthetic datasets quality.. multi-turn settings.."** | **"Missing Graph baselines (KGT)"** **"Missing Routed LLM (RouteLLM)"** | **"Comparison with memory-based methods (MA-GNN).."** **"Comparison with personalized LLM.."** | **"Graph representation vs. Raw interaction.."** | **Summary:** We significantly expanded the evaluation scope. **Reviewer qr28 rebuttal:** We added the LaMP dataset (personalized generation). **Reviewer s59p rebuttal:** We added RouteLLM and MA-GNN as baselines. **Reviewer L1od rebuttal:** We added IPA-RAG (Personalized LLM) and MA-GNN. **Reviewer 8ann rebuttal:** We added TIGER to compare graph-based vs. raw interaction learning. |
| **Scale, Bias & Cost** | **"Scale of Chatbot Arena (11 users) is small.."** **"Validate on scarce/noisy data.."** | "Tests on one real-world and three synthetic benchmarks" | **"Lacks analysis of graph construction cost.."** | "Inductive training... addresses cold-start" **"Multi-user graph clarification.."** | **Reviewer qr28 rebuttal:** We scaled Chatbot Arena experiments to 161 users and conducted noise robustness tests (up to 20% noise). **Reviewer L1od rebuttal:** We provided a detailed table on graph encoding and construction time. **Reviewer 8ann rebuttal:** We clarified that per-user subgraph sampling ensures user independence. |
| **Performance** | "Outperforms existing router baselines" "Efficient, minimal computing" | "Consistently outperforms baselines" "Generalizes well to unseen users" | "Consistently outperforms baselines" "Adapts to new users with minimal data" | "Consistent improvements over baselines" "Lightweight and practical" | **Summary:** All reviewers acknowledge GMTRouter consistently outperforms baselines, is computationally efficient, and generalizes well to new users. |
| **Presentation** | **"Reduce the margin.. formatting.."** | **"Writing is a little bit confusing (selection).."** | "Presentation: Good" | **"Duplicated references.."** **"AUC not defined"** | **Summary:** Most reviewers find the presentation clear. **Reviewer s59p rebuttal:** We revised Section 3 to clarify the link between GNN construction and routing selection. **Reviewer 8ann rebuttal:** We fixed all reference duplications and defined AUC. |

In addition, we expand several minor discussions: such as the specific mechanisms of user-conditioned subgraph sampling, the synergistic effect of combining routing with personalized generation (GMTRouter + IPA-RAG), and detailed runtime analysis. These clarifications do not affect the core contribution of the paper and are fully addressed in the rebuttal.

---

### Note · Authors · 2026-01-06

I have read and agree with the venue's withdrawal policy on behalf of myself and my co-authors.